# Sanitation in Rural India: Exploring the Associations between Dwelling Space and Household Latrine Ownership

**DOI:** 10.3390/ijerph16050734

**Published:** 2019-02-28

**Authors:** Anoop Jain, Lia C.H. Fernald, Kirk R. Smith, S.V. Subramanian

**Affiliations:** 1U.C. Berkeley, School of Public Health, Berkeley, CA 94720, USA; fernald@berkeley.edu (L.C.H.F.); krksmith@berkeley.edu (K.R.S.); 2Harvard T.H Chan School of Public Health, Boston, MA 02115, USA; svsubram@hsph.harvard.edu

**Keywords:** sanitation, social determinants, India, environmental health, sustainable development goals

## Abstract

In 2017, the Joint Monitoring Programme estimated that 520 million people in India were defecating in the open every day. This is despite efforts made by the government, Non-Governmental Organizations (NGOs), and multilaterals to improve latrine coverage throughout India. We hypothesize that this might be because current interventions focus mostly on individual-level determinants, such as attitudes and beliefs, instead of considering all possible social determinants of latrine ownership. Given this, we ask two questions: what is the association between the amount of dwelling space owned by households in rural India and their likelihood of toilet ownership and what proportion of the variation in household latrine ownership is attributable to villages and states? We used multilevel modeling and found significant associations between the amount of household dwelling space and the likelihood of latrine ownership. Furthermore, considerable variation in latrine ownership is attributable to villages and states, suggesting that additional research is required to elucidate the contextual effects of villages and states on household latrine ownership. Thus, sanitation interventions should consider household dwelling space and village and state context as important social determinants of latrine ownership in rural India. Doing so could bolster progress towards Sustainable Development Goal (SDG) 6.

## 1. Introduction

In 2010, the UN General Assembly established access to safe drinking water and toilets as basic rights as they are essential for the “full enjoyment of life and all human rights” [1]. This resolution was adopted because inadequate access to toilets can lead to open defecation, resulting in the spread of fecal contamination throughout the environment. If left untreated, pathogens from this contamination can spread diarrheal disease, the second leading cause of death in children worldwide aged 1–59 months in 2015 [2]. Fecal contamination can also lead to urinary tract infections, soil-transmitted helminth infections, trachoma, cholera, and schistosomiasis [3], and is associated with stunting, a measure of linear growth retardation that can be used as a proxy for economic or educational outcomes [4]. Inadequate access to sanitation also exposes women and girls to unsafe, and sometimes violent, situations [5], while also making menstrual hygiene management extremely difficult. Women in India, for example, have reported withholding food and water in order to limit the number of times they might have to urinate or defecate either during the day or at night [6], while women in Kenya have noted an increase in psychosocial stress associated with not having access to a toilet [7]. Thus, toilets are necessary as a means to prevent infectious diseases, and can also help ensure physical and mental well-being. 

In India, a lower middle-income country with a gross domestic product per capita of $1940 in 2017 [8], approximately 520 million people (almost 40% of the total population) do not have access to any kind of toilet, and thus defecate in the open [9]. Open defecation accounts for nearly 2.5% of the national burden of disease in India, expressed as Disability Adjusted Life-Years according to the Institute for Health Metrics and Evaluation [10], and is estimated to cost India $54 billion annually due to health care costs, losses in productivity, and losses in tourism [11]. 

The Government of India has engaged in efforts to improve toilet coverage over the past several decades. For example, the Central Rural Sanitation Programme, which was founded in 1986, worked with state governments to provide subsidies for individual household latrine construction throughout India [12]. This nation-wide program became the Total Sanitation Campaign in 1999, which evolved into Swachh Bharat Abhiyan (SBA) in 2014. 

Behavior-change curricula, such as Community-Led Total Sanitation (CLTS), have also tried to spur demand for toilet construction by raising awareness about the negative consequences of open defecation [11,12]. CLTS has been implemented throughout India, in addition to South Asia, East Asia Africa, Latin America, and the Caribbean [11,12]. Families living in communities exposed to CLTS are thus supposed to encourage one another to construct and use toilets [11]. However, there is inconclusive evidence that behavior change curricula such as CLTS are effective in encouraging toilet construction [13]. Additionally, CLTS is primarily underpinned by theoretical frameworks designed to motivate individual-level health behavior change, and thus might not account for a broader set of contextual determinants of toilet ownership and use [14]. For example, CLTS has not historically considered the role of gender as a determinant of toilet ownership or use in India [15]. 

A growing body of literature has started examining the possible social determinants of latrine ownership and use in places such as India. Social determinants can be defined as “…specific features of and pathways by which societal conditions affect health and that potentially can be altered by informed action” [16]. In other words, social context is thought to influence various health behaviors and outcomes. For example, Novotny et al. conclude that sanitation change will not be achieved “through specific interventions alone without addressing structural constraints related to educational, economic, and sociocultural inequalities” [17]. Coffey et al. examined one such sociocultural inequality, India’s deeply entrenched caste system, which might perpetuate open defecation. They found that the notions of untouchability that stem from India’s caste hierarchy deter people from using their pit latrines as they do not want to clean them out when they fill up [18]. 

Housing characteristics are also considered social determinants of health outcomes [19], and might be associated with sanitation outcomes. For example, recent studies in Uzbekistan and China suggest that certain housing characteristics, such as access to a centralized water supply, are positively associated with improved water-borne illness outcomes and improved sanitation coverage rates [20,21]. Another critical housing characteristic, the amount of dwelling space owned by a family, is important because the toilet design recommended by the Government of India requires 67 ft^2^ of land. The World Health Organization (WHO) issued Guidelines for Healthy Housing in 1988, which noted the importance of specifying residential density norms so that households would have enough space for a clean latrine to ensure good environmental conditions and hygiene [22]. The report notes that access to sanitation is less likely for those families that live on insufficient amounts of land as they do not have enough space for building a toilet and managing waste [22].

Dwipayanti identified various other social determinants that might be associated with poor sanitation outcomes in Bali, Indonesia [23]. These include poor collaboration and a lack of capacity amongst local government agencies responsible for improving sanitation coverage; village-level economic and social conditions; and a misallocation of responsibility amongst the various agencies responsible for sanitation [23]. Additionally, 13% of the variation in household-level poverty in India is attributable state-level factors, while 12% is attributable to village-level factors [24]. Thus, state and village variation in household poverty, along with the social determinants identified by Dwipayanti, suggest that contextual factors at both the village and state levels might be associated with poor sanitation outcomes in India. 

Thus, the purpose of this paper is to analyze data from the 69th round of the National Sample Survey in India to (1) elucidate the associations between the amount of dwelling space owned by a household and latrine ownership, and (2) examine what proportion of the variation in household latrine ownership is attributable to village-level or state-level factors, using a three-level multilevel analysis (household, village, state). Our hypotheses were that (1) households with larger amounts of dwelling space are more likely to own a toilet than those households with smaller amounts of dwelling space, and (2) there is village-level and state-level variation in toilet ownership.

## 2. Methods

### 2.1. Data

#### 2.1.1. Sampling Strategy

We used the 69th round of the National Sample Survey (NSS) in India, which took place between July and December 2012, to access data about latrine ownership and access [25]. These data were made available by the National Sample Survey Office (NSSO) at the Ministry of Statistics and Program Implementation in India. This survey used a stratified multi-stage sampling design to determine household-level access to drinking water, sanitation, and housing characteristics, as described in the survey report published by the NSSO [26]. We restricted the analyses to rural areas in all states and Union Territories. A full description of how villages were selected can be found in the NSSO report [26]. 

#### 2.1.2. Sample Size and Outcome 

We restricted our analyses to rural India, where the majority of households that lack access to a toilet are located [9]. Overall, the survey captured data from 53,361 rural households located in 4453 rural villages. Latrine access in the survey was divided into five categories: (1) exclusive use by household, (2) common use by households in a building, (3) public/community latrine without payment, (4) public/community latrine with payment, and (5) no latrine. For the purposes of this paper, we only considered categories (1) and (5). This is because the World Health Organization’s Joint Monitoring Programme defines improved sanitation as those “facilities that are designed to hygienically separate excreta from human contact, and that are not shared with other households” [9]. There are several reasons why shared sanitation is not considered improved. For example, some studies suggest that shared sanitation facilities are harder to maintain, leading to unhygienic conditions, which could deter consistent toilet use [27,28,29,30,31]. Thus, we analyzed data from 48,793 households located in 4432 villages after restricting the sample to only those households either with an exclusive household toilet or no toilet at all. 

### 2.2. Measures

#### 2.2.1. Independent Variables

**Primary predictor variable:** Amount of dwelling land owned by a household was divided by 100 to look at the association between every 100 ft^2^ increase in dwelling space and household latrine ownership. We also included a squared term for dwelling space owned by a household as the association between dwelling space owned and latrine ownership might not be linear. 

**Covariates:** We included total monthly expenditure (divided by 250 Rupees to facilitate interpretation), gender of household head (m/f), household head age (years), caste (scheduled tribe, scheduled caste, and other backwards caste), highest educated male in the household, highest educated female in the household, and total number of family members. We also included various household characteristic variables such as dwelling condition (Condition refers to the structure of the dwelling. Enumerators subjectively assessed whether it was good, satisfactory, or bad.) (good, satisfactory, and bad), household electrification, floor type, access to drainage (Drainage refers to how liquid/solid waste is removed from the dwelling. Solid drainage refers to the system being built with concrete, while open means that it was open to the environment without any cover) (underground, covered solid, open solid, open, and no drainage), and roof type. Average amount of dwelling space owned by households in each village and in each state were also included. 

**Interaction Terms:** Our analysis included two interaction terms to test if the association between household dwelling space and toilet ownership is moderated by the average amount of land owned by households in a given village or state. These interaction terms were included because we hypothesized that the strength and direction of the association between the amount of household dwelling space and the likelihood of latrine ownership could be influenced by average household dwelling space by village or by state. 

### 2.3. Analysis

#### 2.3.1. Levels of Analysis

We conducted a three-level analysis in which households (level 1) were nested in villages (level 2), which were nested in states (level 3). We hypothesized that contextual factors at each of the higher levels of analysis could be associated with household (level 1) toilet ownership. 

India underwent large economic and political reform in 1991 [32], during which time states were granted greater autonomy in how policies are implemented. This variation in policy implementation at the state-level highlights the importance of conducting state-level analysis. Variation in economic outcomes could potentially be associated with a variation in household toilet ownership between states, given the significant variation in state-level economic performance and outcomes [33]. 

Villages represent the most similar social, political, and economic environments in which a household could be nested [24]. Thus, there is a need to examine village-level variation in household toilet ownership. 

#### 2.3.2. Analytical Approach

We specified four random-intercept logistic regression models to assess the probability of toilet ownership of household i in village j in state k (y_ijk_ = 1). The four models built on one another in the following way: (1) a fully unadjusted model with only the primary predictor, (2) household-level demographic variables were added, (3) housing characteristics were added, and (4) average household dwelling space by village and state were added. In this final model, we also included two cross-level interaction terms between household dwelling space and the average dwelling space owned by village and state, respectively. 

Each model took the basic form of: logit (π_ijk_) = β_0_ + βX′_ijk_ + (u_0j_ + v_0jk_), where β_0_ is the odds ratio of owning a toilet for a household in the referent category for all of the categorical variables, and when all of the continuous variables are equal to 0, and X′ is the set of variables specified above. We transformed each of the log-odds values to an odds ratio by exponentiating the log-odds result for easier interpretation. In these models, we assume that both random effects (u_0j_ and v_0jk_) are normally distributed with variances of σ^2^_u0j_ and σ^2^_v0jk_, respectively, which signify the between village and between state variations in the odds ratio of latrine ownership, after adjusting for the household level and village level variables. It should be noted that it is not possible to ascertain the level 1 (household) random effect or variance in a logistic regression. Next, we used the variance estimates of the random effects to calculate the variance partitioning coefficient (VPC). This highlighted the proportion of the variation in the log odds of household latrine ownership attributable to the village-level and state-level [34]. We used the latent variable method to calculate the VPC. This method allows for the VPC to be calculated by dividing the variance attributable to a particular level by the total variance. Browne et al. describe the latent variable method and show that it allows for the estimation for the level 1 variance to be π^2^/3 = 3.29 [34]. This value is used given that there is no level 1 variance in logistic regressions. Thus, Browne et al. also show that the variance for level 2, j, is calculated using the following formula: σ^2^_j_/(σ^2^_j_ + σ^2^_z_ + 3.29), where the subscript σ^2^_z_ denotes the level 3 variance [34].

Lastly, we conducted a sensitivity analysis by running a state-level fixed effects model to control for all possible state-level covariates. We did this by including dummy variables for each state into each of our four regression models (description of the models are in the regression results section). We did not include Delhi, Chandigarh, Sikkim, or Lakshadweep as these four Union Territories/States all reported having 100% toilet ownership, and would thus be dropped from the fixed effects model. 

We used Stata 13 SE for descriptive statistics. We used MLwiN 3.00 (University of Bristol, Bristol, United Kingdom) to conduct the multilevel logistic regression analysis for both the three level models and the sensitivity analysis. More specifically, we used iterative generalized least squares (IGLS) to estimate all of the parameters in each of our random effects models and the fixed effects models. 

## 3. Results

We analyzed 48,793 households nested in 4432 villages and in all 28 states and seven union territories in India (There are seven union territories in India. These areas are controlled directly by the federal (national) government. It should also be noted that there were 28 states at the time of this survey (2012), but there are now 29.). The largest sample of villages and households was in Uttar Pradesh with 4914 households in 606 villages, while Dadra and Nagar Haveli was the smallest third-level territory with seven villages and 55 households. Jharkhand and Odisha had the lowest percentage of household latrine ownership, at 14% and 15%, respectively. Overall, 52.9% of the households in our sample did not have a toilet. Furthermore, 87.1% of the households had a male head, while the largest share of households (40.5%) belonged to the Other Backwards Caste category. Additionally, 55.4% of the households in our sample had between three and six family members. In terms of household characteristics, 53.7% of the houses in our sample did not have access to any form of drainage, but 78.2% were electrified. A full set of descriptive statistics and chi-square test values for all of the covariates we included in our models are shown in the Table 1 below.

Latrine ownership was significantly associated with having a larger household dwelling space (Model 1 odds-ratio: 1.35, 95% CI: 1.34, 1.37) (Table 2). This finding was sustained with the inclusion of monthly household expenditure and age of household head, both of which were associated with significantly higher odds of latrine ownership (Model 2 odds-ratio: 1.53, 95% CI: 1.49, 1.57). The findings were also significant after controlling for housing characteristics (Model 3, odds-ratio: 1.35, 95% CI: 1.31, 1.38) and when controlling for household-level socioeconomic variables, housing conditions, and the average amount of space owned by households by village and state (Model 4, odds-ratio: 1.15, 95% CI: 1.09, 1.22).

In examining the other covariates included in the analytic models, we found that higher odds of latrine ownership were significantly associated with having a higher monthly household expenditure and older heads of household, and these associations remain consistent throughout models 2, 3, and 4. Conversely, we found that significantly lower odds of latrine ownership were associated with being of a certain caste (Scheduled Tribe, Scheduled Caste, and Other Backwards Caste households), as well as being in a household in which the highest educated man or woman had less than a college degree. 

There were higher odds of latrine ownership among households with electricity, and those classified as either in good or satisfactory condition compared with those without electricity or in bad condition. Lower odds of latrine ownership were associated with living in households with mud floors (compared with solid floors), open-solid drainage, open drainage, no drainage (compared with underground drainage), and with greater than three family members (compared with fewer than three family members). We found that the odds associated with latrine ownership increased significantly as the average amount of land owned by households in a village or state increased (by 100 ft^2^). 

Lastly, we interpreted the interaction terms and found that the average amount of dwelling space owned by households in a village or state significantly moderates the relationship between the amount of dwelling space owned by a household and the likelihood of latrine ownership (Model 4, odds-ratio: 0.99, 95% CI 0.99, 1.00 & odds-ratio: 1.03, 95% CI 1.02, 1.04, respectively). Thus, the odds of latrine ownership associated with the amount of dwelling space owned by a household increased significantly when the average amount of land owned by households in a village or state increased (by 0.99 and 1.03, respectively). 

We conducted sensitivity analysis by running a state-level fixed effects model that controlled for all state-level variables that might be associated with latrine ownership. We re-did the analysis for models 1 and 4 described above by including dummy variables for each state. 

The results of this analysis, presented in the Appendix A, Table A1, show that higher odds of latrine ownership were significantly associated with larger household dwelling space. Thus, even after controlling for state-level variables that might have been also associated with the odds of latrine ownership, the same household-level covariates remain significantly associated with the odds of latrine ownership. 

### 3.1. Partitioning the Variance

In model 1, we found that 11.1% of the variation in latrine ownership is attributable to the village-level, while 57.7% is attributable to the state-level. Furthermore, we found that there is not much change in the proportion of variance attributable to the village- and state-levels when comparing models 1, 2, and 3 (Appendix A, Table A2). For example, in model 3, we found that 11.2% of the variation in latrine ownership is attributable to villages, while 58.9% of the variation is attributable to states. The total amount of variance attributable to the village- and state-levels decreases in model 4, however, after we included covariates at each of these levels.

## 4. Discussion

Our key findings were that the amount of dwelling space owned by households was significantly associated with latrine ownership after adjusting for all household-level covariates and after adjusting for the average amount of dwelling space at the village and state levels. This association remained significant even in our fixed effects model in which we controlled for all state-level covariates. Furthermore, our finding that the average amount of dwelling space owned by households in a village or state significantly moderates the aforementioned relationship could be reflective of the effects of community-wide crowding or density on the likelihood of latrine ownership. Lastly, 11.1% and 11.2% of the variation in household latrine ownership is attributable to villages, while 57.7% and 58.9% of the variation in household latrine ownership is attributable to states (in our fully unadjusted and adjusted models, respectively). This is possibly indicative of some contextual factors at the village- and state-levels that are associated with household latrine ownership.

There are several limitations to the current study. First, the amount of land owned by a family is often considered a proxy for wealth [35], and thus it might not be a true predictor as we hypothesize it to be in this paper. We mitigated for this by including household-level socioeconomic covariates and housing characteristics, which would account for the “wealth effect”, but may not have accounted for it fully. Additionally, the unit of measurement for area varies regionally across India. Thus, respondents and enumerators might not have accurately captured exactly how much dwelling space a given household owned. Next, these data were initially collected in 2012. While not outdated, they were collected almost six years ago. The number of toilets that have been built since then has increased significantly under SBA, which is not reflected in this analysis. The survey also did not capture certain key demographic information, such as the number of individuals by age in a household. Furthermore, while we used 2nd-order Predictive Quasi-Likelihood (2PQL) to estimate the parameters in our random effects models, we used 1st-order PQL (1PQL) to estimate our parameters in our fixed effects model due to constraints within MLwiN. This could impact the parameter estimates as estimates from 1PQL are often biased downwards [24]. 

Despite these possible limitations, these findings, which quantify the association between dwelling space and the likelihood of latrine ownership, could have several implications. For example, in states such as Bihar, a latrine, which is 67 ft^2^, would take up 18.1% of a household’s dwelling space. Families might be deterred from investing in this infrastructure given that it takes up so much space, especially considering that most Indian households in the sample we analyzed have between three and six family members, which increases crowding and further decreases the amount of available space. 

Additionally, these findings could further the claims that programs such as SBA and CLTS do not do enough to help families actually gain access to a toilet. While SBA was designed to provide financial assistance for latrine construction, no aspect of the policy was designed to account for space constraints that might prohibit a family’s ability to build a toilet. Similarly, CLTS hoped to trigger demand for toilet construction by first spurring awareness about the importance of sanitation. Yet focusing on altering individual attitudes about the importance of sanitation cannot help families overcome space constraints. Thus, these findings could be used to suggest that other types of sanitation be considered as policy makers and governments seek to end open defecation. For example, while individual household latrines (exclusively for one family) are considered the gold standard, shared sanitation facilities could be a viable option for those households and communities that have insufficient space, something the World Bank acknowledges [36]. 

Our results show that a considerable proportion of variation in household latrine ownership is attributable to both village and state conditions. This remains true even after controlling for the average amount of land owned by households per village and state. This suggests that there are possible village-level and state-level contextual factors that are associated with household latrine ownership. For example, Shakya et al. found that social cohesion was a predictor of latrine ownership, a community-wide effect on toilet ownership [37]. Corruption at the village/state levels could also impact latrine ownership outcomes. Indeed, a wide range of corrupt practices in sanitation service delivery were found throughout India [38]. Yet neither social cohesion nor corruption were focuses of this paper, and more research would need to be done to investigate the association between these village- and state-level factors on latrine ownership. 

Lastly, only 5.7% of the households in our sample had access to underground drainage. In fact, 53.7% of the households in our sample did not have access to any form of drainage. This leaves many families without an option for waste management. Our analysis reveals that the estimated odds of household latrine ownership for those with access to underground drainage are 2.7, as much as for households without any drainage (odds-ratio: 0.35, 95% CI: 0.30, 0.40). Having underground drainage could be an indicator of greater household wealth, which in and of itself could be a predictor of latrine ownership. However, we control for household socio-economic variables to help account for this effect. 

The absence of adequate waste management options could deter family latrine ownership for another reason. That is, the Indian government’s recommended sanitation technology in rural communities is the pit latrine. Waste is stored in underground pits, which once full, need to be manually emptied. The ritual impurity associated with this task—stemming from India’s ancient caste system—could serve as a deterrent to toilet ownership or use [3]. 

## 5. Conclusions

Despite decades of government-led sanitation interventions, approximately 520 million people continue defecating in the open throughout India [9]. Our findings, that there exists a significant association between the amount of dwelling space owned by a family and the odds of latrine ownership, could explain why rates of latrine ownership remain low in India, where the average rural household owns less than 500 ft^2^. We also found that this association could be moderated by the mean amount of household dwelling space owned at the village and state levels, which could be indicative of the association between over-crowding and latrine ownership. Lastly, variation in latrine ownership attributable to both villages and states underscores the need for further investigation into various factors that could also be associated with latrine ownership. For example, further research might be needed to understand the associations between state-specific sanitation policies, social cohesion, and corruption on latrine ownership. Overall, our findings suggest that sanitation interventions should consider these, and other, social determinants as a way to bolster India’s progress towards achieving universal sanitation coverage.

## Figures and Tables

**Table 1 ijerph-16-00734-t001:** Descriptive statistics and chi-square test values.

Variable	Household Latrine Access		Percent with Latrine	Chi-Square Test for Independence	
	No	Yes			
Household Dwelling Space Tertiles (sq ft)					
≤270	14,349	2155	13%	χ^2^ (2) = 9300	
>270 and ≤629	12,740	9512	43%	*p* = 0.000	
>629	2,844	7193	72%		
Monthly Household Expenditure Tertiles (Rupees)					
≤3435	11,956	2608	18%	χ^2^ (2) = 6700	
>3435 and ≤7000	15,078	9431	38%	*p* = 0.000	
>7000	2899	6821	70%		
Household Head Gender					
Female	3887	2389	38%	χ^2^ (1) = 1.05	
Male	26,046	16,471	39%	*p* = 0.31	
Household Head Age (years)					
Below 18	259	134	34%	χ^2^ (1) = 3.47	
Above 18	29,674	18,726	39%	*p* = 0.063	
Caste Groups					
Scheduled Caste	4416	3967	47%	χ^2^ (3) = 3600	
Scheduled Tribe	7741	2296	23%	*p* = 0.000	
Other Backwards Caste	13,425	6316	32%		
Other	4351	6281	59%		
Male Formal Education					
No Formal Education	5321	743	12%	χ^2^(4) = 4200	
Literate w/o School, or Below Primary	2816	890	24%	*p* = 0.000	
Primary and Upper Primary	12,043	6165	34%		
Secondary and Higher Secondary	6970	7328	51%		
Diploma and Above	2783	3734	57%		
Female Formal Education					
No Formal Education	10,844	1863	15%	χ^2^ (4) = 6400	
Literate w/o School, or Below Primary	3426	1250	27%	*p* = 0.000	
Primary and Upper Primary	10,419	7460	42%		
Secondary and Higher Secondary	4157	6329	60%		
Diploma and Above	1087	1958	64%		
Housing Condition					
Good	5953	9072	60%	χ^2^ (2) = 5000	
Satisfactory	15,531	7896	34%	*p* = 0.000	
Bad	8449	1892	18%		
Floor Type					
Mud Floor	21,089	5904	22%	χ^2^ (1) = 7200	
Other	8844	12,956	59%	*p* = 0.000	
Household Drainage					
Underground	1145	1649	59%	χ^2^ (4) = 2200	
Covered Solid	950	1260	57%	*p* = 0.000	
Open Solid	3428	3769	52%		
Open	6086	4260	41%		
No Drainage	18,324	7922	30%		
Household Electricity					
Not Electrified	9110	1537	14%	χ^2^ (1) = 3400	
Electrified	20,823	17,323	45%	*p* = 0.000	
Household Size (# of people)					
≤3	8749	4374	33%	χ^2^ (2) = 219.1	
>3 and ≤6	15,872	10,708	40%	*p* = 0.000	
>7	5312	3778	42%		
Roof Type					
Grass/Leaves/Straw/Bamboo	8764	2123	20%	χ^2^ (6) = 5800	
Timber	1702	436	20%	*p* = 0.000	
Burnt Brick/Stone	7530	2755	27%		
Iron/Metal Sheet	2479	1075	30%		
Cement	4667	6336	58%		
Other Solid	4501	5903	57%		
Other	290	232	44%		

**Table 2 ijerph-16-00734-t002:** Regression results (95% CI of OR in parentheses). *** *p* < 0.01, ** *p* < 0.05, * *p* < 0.1.

Response	Model 1	Model 2	Model 3	Model 4
Fixed Part								
Constant	0.58	(0.25, 1.33)	1.26	(0.5, 3.16)	2.36*	(0.90, 6.20)	0.06 **	(0.00, 0.92)
Total Dwelling Space (per 100 ft^2^)	1.35 ***	(1.34, 1.37)	1.53 ***	(1.49, 1.57)	1.35 ***	(1.31, 1.38)	1.15 ***	(1.09, 1.22)
Total Dwelling Space (per 100 ft^2^) Squared			0.99 ***	(0.99, 0.99)	0.99 ***	(0.991, 0.993)	1.00 ***	(0.99, 1.00)
Total Monthly Expense (per 250 Rupees)			1.01 ***	(1.008, 1.013)	1.01 ***	(1.00, 1.01)	1.01 ***	(1.00, 1.01)
Household Head Gender			1.10 **	(1.01, 1.20)	1.13 **	(1.03, 1.25)	1.10 **	(1.01, 1.20)
HH Head Age			1.08	(0.92, 1.27)	1.12	(0.95, 1.34)	1.10	(0.94, 1.29)
Scheduled Tribe			0.39 ***	(0.34, 0.45)	0.52 ***	(0.46, 0.60)	0.56 ***	(0.49, 0.63)
Scheduled Caste			0.49 ***	(0.44, 0.54)	0.57 ***	(0.52, 0.63)	0.63 ***	(0.58, 0.69)
Other Backwards Caste			0.66 ***	(0.60, 0.71)	0.71 ***	(0.65, 0.78)	0.75 ***	(0.69, 0.81)
Male Ed: Illiterate			0.45 ***	(0.40, 0.51)	0.56 ***	(0.49, 0.64)	0.63 ***	(0.56, 0.71)
Male Ed: Literate w/o School or Below Primary			0.51 ***	(0.44, 0.58)	0.63 ***	(0.54, 0.73)	0.68 ***	(0.60, 0.78)
Male Ed: Primary and Upper Primary			0.54 ***	(0.49, 0.59)	0.63 ***	(0.57, 0.70)	0.68 ***	(0.62, 0.75)
Male Ed: Secondary and Higher Secondary			0.73 ***	(0.67, 0.80)	0.78 ***	(0.70, 0.86)	0.81 ***	(0.74, 0.89)
Female Ed: Illiterate			0.43 ***	(0.38, 0.49)	0.53 ***	(0.46, 0.61)	0.59 ***	(0.52, 0.67)
Female Ed: Literate w/o School or Below Primary			0.48 ***	(0.41, 0.55)	0.61 ***	(0.52, 0.72)	0.66 ***	(0.57, 0.77)
Female Ed: Primary and Upper Primary			0.57 ***	(0.51, 0.65)	0.69 ***	(0.60, 0.79)	0.73 ***	(0.65, 0.83)
Female Ed: Secondary and Higher Secondary			0.76 ***	(0.67, 0.86)	0.84 **	(0.73, 0.96)	0.87 **	(0.77, 0.99)
Mud Floor					0.50 ***	(0.46, 0.54)	0.55 ***	(0.51, 0.59)
Condition: Good					1.81 ***	(1.61, 2.02)	1.59 ***	(1.44, 1.77)
Condition: Satisfactory					1.29 ***	(1.17, 1.42)	1.21 ***	(1.11, 1.32)
Drainage: Covered					0.88	(0.73, 1.04)	0.85 *	(0.73, 1.01)
Drainage: Open Solid					0.71 ***	(0.61, 0.82)	0.73 ***	(0.64, 0.83)
Drainage: Open					0.53 ***	(0.46, 0.61)	0.57 ***	(0.49, 0.65)
No Drainage					0.35 ***	(0.30, 0.40)	0.41 ***	(0.36, 0.46)
Electrified HH					1.95 ***	(1.176, 2.16)	1.73 ***	(1.58, 1.89)
HH Size: >3 and ≤6					0.81 ***	(0.75, 0.87)	0.84 ***	(0.78, 0.90)
HH Size: >7					0.67 ***	(0.60, 0.74)	0.70 ***	(0.64, 0.77)
Roof Type: Grass/Leaves/Straw/Bamboo, etc.					0.62 ***	(0.55, 0.69)	0.72 ***	(0.65, 0.80)
Roof Type: Other					0.71 ***	(0.60, 0.85)	0.75 ***	(0.64, 0.88)
Roof Type: Timber					0.77 ***	(0.70, 0.85)	0.81 ***	(0.74, 0.89)
Roof Type: Burnt Brick/Stone					0.66 ***	(0.57, 0.76)	0.72 ***	(0.63, 0.81)
Roof Type: Iron/Metal Sheet					0.74 ***	(0.66, 0.82)	0.78 ***	(0.71, 0.86)
Roof Type: Other Solid					0.86	(0.65, 1.15)	0.90	(0.69, 1.17)
Mean Village Dwelling Space (per 100 ft^2^)							1.05 **	(1.01, 1.09)
Mean State Dwelling Space (per 100 ft^2^)							2.01 **	(1.17, 3.46)
Village HH Dwelling Space Interaction							0.99 ***	(0.99, 1.00)
State HH Dwelling Space Interaction							1.03 ***	(1.02, 1.04)
Random Part								
State Variance	6.07		6.166		6.472		4.323	
Village Variance	1.159		1.171		1.216		0.773

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
