# Peer review of "Sanitation in Rural India: Exploring the Associations between Dwelling Space and Household Latrine Ownership"

_ijerph, 2019, doi:10.3390/ijerph16050734_

Round 1

Reviewer 1 Report

The article deals with important topic which is now at the centre of attention due to the inclusion of sanitation targets to SDGs and particularly in Indie due to the ongoing Swachh Bharat Mission. It is undoubtedly relevant topic for this journal and methods used for the analysis seem to be appropriate for a given analysis. However, I do think that the article should notably be improved and that it is not publishable in its current form. More specific comments are as follows:

Although I am myself not a native English speaker, I was uneasy with the quality of English in this manuscript and the language style in general.

I do not think that the present title well correspond to what is addressed in the analysis. For some reasons that I do not fully understand, the authors present the focus on the role of dwelling space on latrine ownership as one of the two research questions (p. 2-3). But the analysis also examines other factors that may be similarly or more relevant as “social determinants” of sanitation (announced in the title).

In my opinion, the general description of the importance of sanitation in Introduction (p. 1-2) can be reduced and replaced (or supplemented) by more appropriate discussion of evidence on the role of particular factors including social determinants in sanitation with specific focus on India. The insufficient discussion of directly relevant recent literature is a problem of this study.  

In particular, I suggest consulting more recent papers by D. Spears and D. Coffey and their colleagues - https://scholar.google.cz/citations?hl=cs&user=XzPR1u8AAAAJ&view_op=list_works&sortby=pubdate  , https://scholar.google.cz/citations?hl=cs&user=MYcDz1AAAAAJ&view_op=list_works&sortby=pubdate

Novotny et al – 2018 - Social determinants of environmental health: A case of sanitation in rural Jharkhand (https://www.sciencedirect.com/science/article/pii/S0048969718323167 ), and 2017 - Contextual factors and motivations affecting rural community sanitation in low- and middle-income countries: A systematic review https://www.sciencedirect.com/science/article/pii/S143846391730723X

India is lower-middle income country and not low-income country (line 45)

As far as I know, the use of CLTS has neither been prevalent nor much successful in India

compared to some other countries and where it has been used in India it was often with divergences from the CLTS principles such as a no subsidy principle. The references to CLTS in Introduction (p. 2) are not appropriate in my opinion.

Moreover, the CLTS is an approach and not a program that can be compared to SBA as it is on the p. 2.

Moreover, if you check the CLTS handbook and SBA guidelines in both these documents you can find numerous references to the importance of the context, needs for local adjustments of the sanitation interventions etc. The problem however is that this typically remains on paper only. However, the statement on the line 84 that there is no sensitiveness to context is not true.

Generally, I think that these paragraphs on the p. 2 should be revised carefully.   

Consider the extension of the first research question to account for other factors analyzed in the paper too. Otherwise, title should be changed and the introductory discussion should be more focused on the role of your main factor (dwelling space).

In my opinion, the structure of the Method section is confusing. I suggest dividing this section into three sub-sections including Data, Measures, and Analytical approach sections. You use secondary data so that the description of sampling frame including sample size can be in Data section, description of measures/variables in the Measures section, and analytical procedure including the levels of analysis can be outlined in the Analytical Approach section.

The problems of data reliability should be mentioned and discussed (if possible).

The current version of the text in the Sampling strategy section is unclear (lines 108-114).

In my experience, district level is quite important for sanitation policy implementation in India. Why the district level is not considered in the multi-level analysis in addition to village and state level?

The analysis seems correct to me but I would like to know more about possible interrelations between dwelling space and other socioeconomic variables. As mentioned in the paper, there is a risk of false correlation (so called “wealth effect” mentioned on the line 270).

Again, it is not clear in the interpretation of results why the dwelling space is taken as a focal variable and other factors as covariates only (particularly if Introduction provides no theoretical substantiation for specifically the role of dwelling space). The discussion of results can be considerably richer, taking also other variables into consideration and confronting the results to evidence from other relevant literature. The absence of more elaborate discussion makes the paper descriptive in character.    

Author Response

See attached pdf file

Reviewer 2 Report

General comments:

A well conceived and written study. As a suggestion, it may be useful to consider introducing and elucidating the term 'structural determinants' in the discussion on the variation of latrine ownership associated with village and state level factors (e.g., from Dwipayanti, 2017: policy and regulation for domestic wastewater management; allocated responsibility and accountability for action; degree of capacity and degree of collaboration of local government institutions; village social and organisational structures and economic status; status of sanitation curriculum in the education system). This could also be appropriate in your conclusions, adding more specificity when characterising the call for further research into 'the associations between state-specific sanitation policies, social cohesion, and corruption on latrine ownership'.

Specific comments:

page 2, line 47:    Suggest reword the sentence beginning "Open defecation..." to clarify its meaning. e.g. "Open defecation accounts for nearly 2.5% of the national Burden of Disease in India, expressed as Disability-Adjusted Life Years, according..."

e.g. page 6, line 223: Check throughout the paper for consistency in the use of a plural verb form with the plural noun 'odds'

Author Response

See attached pdf file
